Biostimulant effects of titanium dioxide nanoparticles on germination and initial growth of tomato: evidence of hormesis

Carbajal-Vázquez Víctor Hugo 1
Trejo-Téllez Libia Iris 2 3
Mejía-Méndez Jorge Luis 4
Salinas-Ruiz Josafhat 5
Gómez-Merino Fernando Carlos fernandg@colpos.mx 1
1 Department of Plant Physiology, College of Postgraduates in Agricultural Sciences , Montecillo, Texcoco , State of Mexico , Mexico
2 Department of Soil Science, College of Postgraduates in Agricultural Sciences , Montecillo, Texcoco , State of Mexico , Mexico
3 Department of Plant Physiology, College of Postgraduates in Agricultural Sciences Montecillo Campus , Texcoco , State of Mexico , Mexico
4 Department of Soil Science, College of Postgraduates in Agricultural Sciences , Texcoco , State of Mexico , Mexico
5 Department of Sustainable Agri-food Innovation, College of Postgraduates in Agricultural Sciences , Amatlán de los Reyes , Veracruz , Mexico
Brygadyrenko Viktor
Electronic publication date: 2025 Dec 16
Publication date: 2025
Volume: 13
Electronic Location ID: e20516
Received 2025 Mar 5; Accepted 2025 Nov 12
Copyright: ©2025 Carbajal-Vázquez et al.
Copyright year: 2025
Copyright holder: Carbajal-Vázquez et al.
License: This is an open access article distributed under the terms of the Creative Commons Attribution License, which permits unrestricted use, distribution, reproduction and adaptation in any medium and for any purpose provided that it is properly attributed. For attribution, the original author(s), title, publication source (PeerJ) and either DOI or URL of the article must be cited.
License URL: https://creativecommons.org/licenses/by/4.0/

Keywords: Solanaceae, Solanum lycopersicum, Inorganic biostimulants, Nanoparticles, Imbibition, Hormesis, Dose-response curves

Funding: Mexico’s Secretariat of Science, Humanities, Technology and Innovation SECIHTI; CVU 704278 The work was funded by a postdoctoral scholarship granted to Víctor Hugo Carbajal Vázquez by the Mexico’s Secretariat of Science, Humanities, Technology and Innovation (SECIHTI; CVU 704278). The funders had no role in study design, data collection and analysis, decision to publish, or preparation of the manuscript.

==============================
Background

Titanium (Ti) is a beneficial element considered an inorganic biostimulant that may induce hormesis in plants. Supplied as titanium oxide nanoparticles (nTiO2) at appropriate concentrations, it may enhance seed germination, plant vigor, photosynthesis, and abiotic stress tolerance, while increasing nutrient uptake, crop yield and nutritional value. Nonetheless, applied at high concentrations, nTiO2 may trigger detrimental effects in plants.

Methods

Tomato (Solanum lycopersicum L.) cv. Rio Grande seeds were imbibed in solutions containing 0.0, 52.2, 104.4, 156.6, or 208.8 µM nTiO2. The germination experiment lasted 16 days under controlled conditions in the laboratory (10 h natural light, 21 °C mean temperature, at saturated atmospheric conditions). During the experiment we estimated seed weight increase after imbibition; total germination percentage (TGP); germination speed coefficient (GSC); and the vigor indexes I and II (VI I and II). Seedlings were grown for 30 days after sowing and at the end we measured lengths of stems and roots, number of roots and leaves, moisture content, and fresh and dry biomass weight. In addition, the potential hormetic effect of Ti on length of roots and stems was estimated.

Results

Applications of 156.6 and 208.8 µM nTiO2 significantly increased vigor index I, root length and total moisture content in leaves, while applying 208.8 µM nTiO2 significantly increased fresh biomass weight of roots. The hormetic analysis revealed that the application of 156.6 µM nTiO2 stimulated the length of roots and stems, with different dose-response curves.

Conclusion

The application of nTiO2 to tomato seeds improved some germination and plant growth parameters during the initial growth stage, demonstrating its biostimulant effects.

Introduction

Nanotechnology is a fast-evolving field focused on the design and manipulation of matter at the nanometric scale. Contrary to their bulk counterparts, nanomaterials can exhibit unique physical, chemical, and biological features. The performance of organic and inorganic nanomaterials relies on their chemical composition, synthesis route, and spectroscopy and microscopy features. Nanomaterials can be synthesized from carbon precursors, polymers, ceramics, and metal oxides.

Titanium (Ti) is a transition metal found at a concentration of 0.57% of the Earth’s crust, ranking ninth among the chemical elements of the periodic table in terms of abundance (Van Gosen & Ellefsen, 2018). Ti is mainly extracted from minerals such as ilmenite, leucoxene, and rutile (Kotova et al., 2016; Maldybayev et al., 2024), with ilmenite providing about 92% of the world’s Ti supply (Bedinger, 2018). The main producing countries of mineral Ti are Australia, South Africa, China, and Mozambique, and global production of this element amounts to more than 10.3 million megagrams (Mg), with annual increases of approximately 4% (Van Gosen & Ellefsen, 2018) because of its broad applications in very diverse industries such as biomedicine, farming and food processing (Pushp, Dasharath & Arati, 2022; Saurabh et al., 2022). Given the expansion of the use of Ti, greater exposure of living beings and the environment to the metal is expected; therefore, more detailed studies aimed at unraveling the effects of Ti on various biological systems and the environment remain a daunting task.

Since population growth represents one of the main challenges currently facing humanity, with an increase of 56% in food demand estimated by 2050 (OECD/FAO, 2024), agriculture needs to develop and adapt technologies to produce food in more limiting environments and reduce the negative impact of stress factors of both a biotic and abiotic nature. Consequently, in recent years, the use of nanotechnology has taken on an important role in agriculture as a tool to improve crop production systems, allowing the controlled release of nanoagrochemicals in order to enhance productivity and food supply (Chaud et al., 2021; van Dijk et al., 2021; Santás-Miguel et al., 2023; Rodríguez-Seijo et al., 2025).

Titanium oxide nanoparticles (nTiO2) have anticorrosive and photocatalytic properties (Heckman et al., 2024; Mao & Hao, 2024), which are currently used in food additives (E171), or as an excipient in medicines, as long as it does not exceed 1% by weight of the product, as stated by the US Food and Drug Administration (FDA regulations 21 CFR 73.575). With growing demand, the current production of nTiO2 isclose to 88,000 Mg year−1, making it the nanomaterial with the greatest release into the environment. In Europe, an input of 0.13 µg nTiO2 kg−1 of soil yearly is estimated, with the potential to increase to 1,200 µg kg−1 year−1 due to waste sludge discharges (Abukabda et al., 2017; Tiwari et al., 2017; Radziwill-Bienkowska et al., 2018).

In the agricultural and food sectors, the use of metals in the form of nanoparticles is very common due to the various properties that they provide, which contributes to their accumulation in waste sludge whose effects on the biota and the environment are not yet known in detail (Jan et al., 2022; Tran et al., 2024).

In higher plants, the potential positive, neutral, or negative effects of bulk or nano-Ti are mediated by processes of hormesis, a dose–response relationship conditioned by environmental factors, which at low doses stimulates biological processes, but inhibits them at high doses. This natural phenomenon represents an evolutionary strategy of the species, restricted by biological plasticity that allows adaptive responses to environmental challenges (Calabrese, 2021). Hormetic responses are considered normal within the physiological functions of an organism at the cellular level, which allows a pre-conditioning to a specific environment. The control of hormetic responses in plants can contribute to improving growth, development, and yield attributes in agricultural systems (Agathokleous & Calabrese, 2019; Jalal et al., 2021).

In plant biology, Ti is considered a beneficial element that can trigger hormetic responses when applied as bulk material or as a nanoparticle (Lyu et al., 2017; Qi et al., 2024). In spinach (Spinacia oleracea L.), the application of nTiO2 increased the enzymatic activities of catalase (CAT), superoxide dismutase (SOD), and peroxidase (POX), and decreased the levels of malondialdehyde (MDA) and that of reactive oxygen species (ROS), while keeping the stability of plastid membranes under high light conditions (Hong et al., 2005) and stimulating the activity of key enzymes involved in N metabolism (i.e., glutamate dehydrogenase, glutamine synthase, glutamate-pyruvate transaminase and nitrate reductase) (Fan et al., 2006). In wheat (Triticum aestivum L.), Ti reduced the mean germination time and improved seedling development (Feizi et al., 2012; Ates-Sonmezoglu et al., 2024). In fennel (Foeniculum vulgare P. Mill.), nTiO2 enhanced seed germination percentage and seedling growth rate (Feizi et al., 2013). In tomato (Solanum lycopersicum L.), the application of a nTiO2 (80:20 anatase:rutile), with a 27 nm particle size, had neither toxic nor stimulatory effects on seeds or plants (Song et al., 2013). In onion (Allium cepa L.), nTiO2 induced the germination of seeds and the growth of seedlings, in addition to inducing the activity of hydrolytic and antioxidant enzymes (Laware & Raskar, 2014). In mung bean (Vigna radiata [L.] R. Wilczek), nTiO2 affected the germination mechanism and grain growth rate (Mathew, Sunny & Shanmugam, 2021). In strawberry (Fragaria x ananassa Duch.), nTiO2 (125.2, 250.4, and 375.6 µM; 21 nm in size with 99.5% purity; Evonik; Germany) reduced the negative effects of drought stress while improving productivity (Javan et al., 2024). Similar results were obtained in grapevine (Vitis vinifera L.) (Daler et al., 2024). Just recently, Trela-Makowej, Orzechowska & Szymańska (2024) performed an integrative analysis of the effects of various concentrations of nTiO2 on different plant responses, including photosynthesis, oxidative stress, and regulation of gene expression, concluding that nTiO2 exhibits a complex, multifaceted impact on plant-biology, showing both promise and challenges for applications in agriculture. Nevertheless, little has been explored about the hormetic effect of nanotitanium by performing in-depth mathematical and statistical analyses. Herein, we aimed to evaluate the effect of the application of nTiO2 at five doses (0, 52.2, 104.4, 156.6, and 208.8 µM nTiO2) on seed germination and initial growth of tomato seedlings in order to derive potential hormetic curves. The hypothesis to be tested was that low concentrations of nTiO2 (i.e., 52.2 or 104.4 µM) may have biostimulant effects on the processes of seed germination and initial growth of tomato seedlings, and that the variables evaluated may trigger hormesis, with low dose stimulation and a high dose inhibition.

Materials and Methods

Plant material and treatments

Saladette-type hybrid tomato cv. Rio Grande seeds were provided by Geneseeds (Mexico). Healthy and homogeneous seeds (without observable physical damage) were selected, and then weighed in groups of ten. Subsequently, a stock solution of titanium dioxide nanoparticles (nTiO2, <25 nm in size, 99.7% purity; Sigma-Aldrich, St. Louis, MO, USA) with a concentration of 208.8 µM was prepared. Hence, from the purchased nanoparticles of <25 nm in size we prepared the stock solution at a final concentration of 208.8 µM. We then diluted this stock solution to reach the different concentrations of the treatments tested that are described below.

Dilutions were made from the stock solution to 52.2, 104.4, and 156.6 µM nTiO2; the control solution was distilled water. These dilutions were prepared based on previous studies summarized by Trela-Makowej, Orzechowska & Szymańska (2024). In order to ensure minimal agglomeration, samples were placed onto the platform of a magnetic stirrer (Labnet; Edison, NJ, USA) for 10 min under constant agitation at 300 rpm, prior to applications. The hydrodynamic size, surface charge, and polydispersity index of the diluted samples were determined utilizing a Microtrac Nanotrac Wave II (Montgomeryville, PA, USA) instrument, in accordance with the protocol described by (Mejía-Méndez et al., 2024). For this analysis, samples were constantly agitated and analyzed without further dilutions. All experiments were performed in triplicate.

The seeds underwent an imbibition process for 24 h under controlled laboratory conditions (10 h natural light, 21 °C mean temperature, at saturated atmospheric conditions), following the methodology described by Moret-Fernández, Tormo & Latorre (2024). Accordingly, 10 seeds were placed in glass bottles containing 30 mL of each of the solutions to be evaluated. After imbibition, the seeds were removed from the bottles, dried with absorbent paper, and weighed again. With the difference in weights, the seed weight increase after imbibition (SWIAI) was estimated (ISTA, 2010).

Germination

After seed imbibition for 24 h in the different nTiO2 solutions, groups of 10 seeds were placed in plastic containers with lids (12 × 11 × 7 cm). Each container was provided with an 11 × 10 cm piece of filter paper, in which five mL of distilled water was poured. During the time the experiment was carried out, the moisture of the experimental unit was maintained by adding five mL of distilled water every third day. The experimental unit was represented by a container with 10 seeds. Each experimental unit had three replications. For seed germination measurements, records were taken every 24 h; once data was constant, we stopped taking records and performed the calculations.

Germination kinetics

Germination recordings were made daily for 16 days. Any seed that reached a radicle more than two mm long was considered a germinated seed.

Total germination percentage (TGP). This variable was estimated according to Duclos et al. (2013). The TGP measures the real percentage value of germinated seeds, and considers the maximum germination value reached in the kinetics (constant value), as follows: TGP=SproutedseedsTotalseeds×100.

Germination speed coefficient (GSC). It was estimated according to the following method described by Kader (2005): GSC=Total number of germinated seeds per experimental unitA1T1+A2T2+AXTX

where A1, A2…= The number of seeds sprouted in a particular number of days. T1, T2…is the number of seed germination days after the start of incubation.

Vigor index I and II. They were determined according to the methodology proposed by Vashisth & Nagarajan (2010), using the following formulae:

Vigor index I = (Percentage of germination) × [Seedling length (Root+Shoot)]

Vigor index II = (Percentage of germination) × [Seedling dry biomass weight (Root+Shoot)].

Growth and biomass production

For early growth measurements, seedlings were kept under the same controlled conditions in the laboratory as aforementioned. Thirty-five days after sowing (das), the lengths of the shoot and main root were measured with a 10 cm long graduated ruler. The number of secondary roots was counted manually using a magnifying glass, and the number of leaves was also counted manually. Immediately after having completed these measurements (35 das), individual plants were harvested, and roots, stems and leaves were separated for further analyses. The samples were weighed on an Adventurer Ohaus Pro AV213C analytical balance (Parsippany, NJ, USA) to obtain the weight of fresh biomass. Subsequently, samples were deposited in paper bags to be dried at 70 °C for 48 h in a forced air oven (Riossa HCF-125D; Guadalajara, Jalisco, Mexico). Finally, the weight of the dry biomass was obtained on the analytical balance.

Total moisture content (TMC). It was determined in roots, stems, and leaves, taking into consideration both fresh and dry biomass weight, as follows: TMC%=Fresh biomass weight−Dry biomass weightFresh biomass weight×100.

Hormetic effects

To test whether the shoot and root length variables displayed hormetic dose curves in response to nTiO2, data were fitted to the model developed by Brain & Cousens (1989), which is defined as: Eyij=c+d−c+fxij/1+xij/eb

where yij represents the response in the jth repetition at the ith concentration of nTiO2; Xij is the ith concentration level of nTiO2; c indicates the frequency response at infinite doses; d represents the average response of the untreated control; f and e designate the degree of increase in hormesis (f > 0 as a necessary condition for the presence of hormesis); and b is the size of hormesis. All these statistical analyses were run using the R statistical software (Venables & Smith, 2024) and the dcr library (Ritz et al., 2015).

Experimental design and statistical analysis

In this study we established a completely randomized experimental design (CRD) with five treatments and three replicates each. The experimental unit was a plastic container with 10 seeds. With the results obtained, we first performed an analysis of variance and then a mean comparison test using the LSD method (p ≤ 0.05). For this, SAS software was used (SAS, 2023).

Results

Characterization of the titanium dioxide nanoparticles (nTiO2)

As observed in Fig. 1, the average hydrodynamic size of nTiO2 samples varied according to the performed dilution. For instance, it was noted that the average hydrodynamic size of the control samples (i.e., containing no nTiO2) ranged from 98.6 to 356 nm. The zeta potential (ζ-potential) of this sample was 0.2 mV. On the contrary, the average hydrodynamic size of nTiO2 at 52.2 µM nTiO2 ranged from 417 to 3,040 nm with a ζ-potential of 8.5 mV. Similarly, the hydrodynamic size of nTiO2 at 104.4 µM nTiO2 5 mg varied from 501 to 1,577 nm with a ζ-potential of 8.8 mV. At 156.6 µM nTiO2, the Dynamic Light Scattering (DLS) analysis of nTiO2 revealed three predominant peaks at 521, 778, and 3,850 nm, respectively. The ζ-potential of this sample was 9.1 mV. The average hydrodynamic size of nTiO2 at 208.8 µM nTiO2 was 327 nm with a ζ-potential of 10 mV. Thus, the ζ-potential increased according to the concentration of nTiO2 contained in the sample analyzed.

Figure 1 Dynamic Light Scattering (DLS) analysis of titanium dioxide nanoparticles (nTiO2) applied at different concentrations: (A) 0; (B) 52.2; (C) 104.4; (D) 156.6; and (E) 208.8 μM.

Weight gain in tomato seeds

In the study, nTiO2 did not have significant effects on seed weight increase after imbibition (SWIAI) compared to the control. However, seeds imbibed with 104.4 µM nTiO2 gained 114.5% more weight than seeds imbibed with 208.8 µM nTiO2, and this difference was significant (Fig. 2).

Figure 2 Boxplot of seed weight changes in tomato (Solanum lycopersicum L.) cv. Rio Grande after 24 h of imbibition with different concentrations of titanium dioxide nanoparticles (nTiO2).

Boxes indicate the interquartile range, the line and diamond represent the median and mean, respectively. Different letters show significant differences among treatments (LSD, p ≤ 0.05) n = 3.

Germination variables

As observed in Table 1, the means of the total germination percentage (TGP) were similar among all the treatments tested. Likewise, Ti doses had no effect on the germination speed coefficient (GSC) since all Ti treatments were statistically similar to the control. In the case of vigor indexes I (VI I) and II (VI II), similar trends were observed between them. As for VI I, the seedlings from seeds treated with 156.6 and 208.8 µM nTiO2 showed a higher value compared to the control by 26 and 27%, respectively. For VI II, the application of 208.8 µM nTiO2 resulted in higher means as compared to the application of 104.4 µM nTiO2, and though the highest dose of Ti applied (i.e., 208.8 µM nTiO2) surpassed the control by 21.7%, all means were similar to the control.

Table 1 Biostimulant effects of titanium dioxide nanoparticles (nTiO2) on germination parameters of tomato (Solanum lycopersicum L.) seeds cv. Rio Grande.

Concentrations of nTiO2applied (μM)	Total germination percentage	Germination speed coefficient	Vigor index I	Vigor Index II	
0.0	90.0 ± 8.7a	0.170 ± .006a	1,170.5 ± 120.5b	48.3 ± 8.1ab	
52.2	86.7 ± 2.9a	0.168 ± .005a	1,228.9 ± 49.0ab	46.6 ± 3.8ab	
104.4	80.0 ± 5.0a	0.177 ± .020a	943.1 ± 57.8b	36.6 ± 2.4b	
156.6	90.0 ± 5.0a	0.157 ± .031a	1,476.0 ± 107.9a	51.0 ± 5.1ab	
208.8	96.6 ± 2.9a	0.151 ± .009a	1,485.6 ± 47.6a	61.7 ± 6.8a	
Notes.

Tomato seeds were imbibed in solutions containing different concentrations of titanium dioxide nanoparticles (nTiO2; 25 nm) for 24 h. Germination parameters were recorded for 16 days. Means ± SD with different letters in each variable indicate statistical differences among treatments (LSD, p ≤ 0.05). n = 3.

Root and stem length, number of roots and leaves

The average values of the variables root length, stem length, number of roots and number of leaves are displayed in Table 2. The greatest root length was observed in seedlings of the seeds exposed to 156.6 µM nTiO2, with an increase of almost 31% with respect to the control. Although the stem length means were all statistically similar, the 156.6 µM nTiO2 dose resulted in an increase of 16.3% compared to the control. The treatments applied did not affect the number of lateral or secondary roots. However, the number of leaves decreased by 32.2, 37.1, and 26.6% when applying 52.2, 104.4, and 156.6 µM nTiO2, respectively, compared to the control.

Table 2 Biostimulant effects of titanium dioxide nanoparticles (nTiO2) on growth parameters of tomato (Solanum lycopersicum L.) seedlings cv. Rio Grande.

Concentrations of nTiO2applied (μM)	Root length (cm)	Stem length (cm)	Number of lateral roots	Number of leaves	
0.0	9.40 ± 0.46bc	3.62 ± 0.21a	3.88 ± 0.19a	1.43 ± 0.06a	
52.2	10.41 ± 0.77abc	3.86 ± 0.32a	3.54 ± 0.29a	0.97 ± 0.17bc	
104.4	7.94 ± 0.17c	3.85 ± 0.06a	3.79 ± 0.15a	0.90 ± 0.08c	
156.6	12.30 ± 1.12a	4.21 ± 0.27a	3.35 ± 0.27a	1.05 ± 0.16abc	
208.8	11.44 ± 0.59ab	3.94 ± 0.09a	4.10 ± 0.23a	1.36 ± 0.03ab	
Notes.

Tomato seedlings emerged from seeds imbibed in solutions containing different concentrations titanium dioxide nanoparticles (nTiO2; 25 nm) for 24 h, were grown for 30 days. Means ± SD with different letters in each variable indicate statistical differences among treatments (LSD, p ≤ 0.05). n = 3.

Total moisture content in tomato seedlings

The moisture content in roots, stems, and leaves was differently affected by the Ti treatments tested (Fig. 3). In roots, moisture content increased with the application of 156.6 and 208.8 µM nTiO2, as compared to the control (Fig. 3A), while in stems, seeds treated with 104.4 µM nTiO2 increased moisture content as compared to the control (Fig. 3B). In leaves, the moisture content in seedlings from seeds treated with 208.8 µM nTiO2 was 2.6% higher compared to the control treatment; the application of 104.4 µM nTiO2 also increased the moisture content of seedlings as compared to the control (Fig. 3C).

Figure 3 Boxplots of total moisture content in seedlings of tomato (Solanum lycopersicum L.) cv. Rio Grande exposed to different concentrations of titanium dioxide nanoparticles (nTiO2) 30 days after sowing.

(A) Roots; (B) Stems; (C) Leaves. Boxes indicate the interquartile range, the line and diamond represent the median and mean, respectively. Different letters within each variable denote significant differences among treatments (LSD, p ≤ 0.05). n = 3.

Fresh and dry biomass production

Different responses were found in the means of fresh and dry biomass weight in seedlings from seeds treated with Ti. With the 208.8 µM nTiO2 dose, the greatest amount of fresh biomass was produced in roots, with an average increase of 35.2% compared to the control. The 52.2 and 104.4 µM nTiO2 doses reduced stem fresh biomass by 20.9 and 45.7% compared to the control. In leaves, the application of 208.8 µM nTiO2 increased the fresh biomass weight by 23.9% as compared to the application of 104.4 µM nTiO2. Similar results were observed for the total fresh biomass of the whole seedlings, since seeds imbibed with 208.8 µM nTiO2 resulted in seedlings with 27.8% more fresh biomass than those treated with 104.4 µM nTiO2, while all means were similar to the control (Table 3).

Table 3 Fresh biomass weight of tomato (Solanum lycopersicum L.) seedlings cv.

Rio Grande treated with different concentrations of titanium dioxide nanoparticles (nTiO2).

Concentrations of n TiO 2 applied (μM)	Fresh biomass weight (mg)	
	Roots	Stems	Leaves	Whole seedlings	
0.0	152.4 ± 30.0b	188.4 ± 6.7a	184.8 ± 8.0ab	525.7 ± 44.3ab	
52.2	201.2 ± 8.6ab	155.8 ± 7.8bc	178.5 ± 13.4ab	535.3 ± 29.6ab	
104.4	167.6 ± 6.8ab	129.3 ± 8.4c	161.2 ± 6.5b	458.1 ± 18.7b	
156.6	197.6 ± 20.5ab	181.5 ± 0.9ab	184.9 ± 9.2ab	564.0 ± 29.6ab	
208.8	235.2 ± 31.6a	187.3 ± 12.5a	212.0 ± 13.2a	634.5 ± 53.6a	
Notes.

Tomato seedlings emerged from seeds imbibed in solutions containing different concentrations of titanium dioxide nanoparticles (nTiO2; 25 nm) for 24 h, were grown for 30 days. Means ± SD with different letters in each variable indicate statistical differences among treatments (LSD, p ≤ 0.05). n = 3.

The dry biomass weight of roots was statistically similar among treatments. In stems, the 104.4 µM nTiO2 doses applied to the seed decreased the dry biomass weight of seedlings by 43.2%, compared to the control. Dry biomass weight of leaves decreased by 27.1% when seeds were treated with 104.4 µM nTiO2, as compared to the control. The total dry biomass of the seedlings was 24.2% lower in seedlings from seeds imbibed with 104.4 µM nTiO2 than those treated with 0 µM nTiO2 (Table 4).

Table 4 Dry biomass weight of tomato (Solanum lycopersicum L.) seedlings cv.

Rio Grande treated with different concentrations of titanium dioxide nanoparticles (nTiO2).

Concentrations of n TiO 2 applied (μM)	Dry biomass weight (mg)	
	Roots	Stems	Leaves	Whole seedlings	
0.0	12.0 ± 1.3a	8.8 ± 0.5a	15.5 ± 0.9a	36.3 ± 2.5a	
52.2	13.6 ± 0.7a	7.5 ± 0.4a	14.5 ± 0.8ab	35.7 ± 1.8ab	
104.4	11.2 ± 1.8a	5.0 ± 0.3b	11.3 ± 0.6b	27.5 ± 2.5b	
156.6	11.7 ± 1.4a	7.5 ± 0.4a	14.2 ± 1.0ab	33.3 ± 2.7ab	
208.8	13.7 ± 1.5a	8.1 ± 0.9a	12.4 ± 1.3ab	34.2 ± 2.4ab	
Notes.

Tomato seedlings emerged from seeds imbibed in solutions containing different concentrations of titanium dioxide nanoparticles (nTiO2; 25 nm) for 24 h, were grown for 30 days. Means ± SD with different letters in each variable indicate statistical differences among treatments (LSD, p ≤ 0.05). n = 3.

Hormetic effect

According to our analyses, the application of nTiO2 resulted in hormetic dose–response effects for root length (RL) and shoot length (SL), with a stimulatory behavior at low or intermediate Ti doses, while at high Ti doses it was inhibitory (Fig. 4). The dose–response curves obtained were inverted U-shaped for SL and J-shaped for RL (Kendig, Le & Belcher, 2010). The dose with 104.4 µM nTiO2 remained above the hormetic zone for SL. The dose of 156.6 µM nTiO2 caused the highest stimulation value, while, with 208.8 µM nTiO2, toxicity was observed at the threshold. Regarding RL, it was observed that the dose of 104.4 µM nTiO2 tends to decrease the value of this variable, while the doses of 156.6 and 208.8 tend to increase it.

Figure 4 Hormetic response in the variable root and shoot length in seedlings of tomato (Solanum lycopersicum L.) cv. Rio Grande exposed to different concentrations of titanium dioxide nanoparticles (nTiO2) 30 days after sowing.

(A) Root length: (a) maximum stimulatory response; (b) hormetic zone; (c) and (d) toxic threshold. (B) Shoot length. In all experiments, n = 3.

Discussion

Germination is a fundamental process by which the hydrated embryo develops an axis and culminates in the emergence of a new plant. During this process, the state of latency is broken, allowing respiration, protein synthesis, and mobilization of reserves and germination to be activated, and embryonic growth begins. Seed germination may reach maximum rates when factors such as moisture, temperature, oxygenation, and luminosity are optimal (ISTA, 2005; Ruiz-Nieves et al., 2021), while some imbibition treatments can enhance germination by improving hydration and activating key enzymes in seeds of different species (Monteon-Ojeda et al., 2021; Pompelli, Jarma-Orozco & Rodríguez-Páez, 2023; Upretee, Bandara & Tanino, 2024).

Titanium dioxide nanoparticles (nTiO2) can increase resistance to stress and promote water penetration to the seed, which causes greater weight gain (Mustafa et al., 2021). The effects of nTiO2 can be influenced by factors such as type of nanoparticle, dose, form of application, exposure time, and genotype of plant to be treated (Acosta-Slane et al., 2024; Trela-Makowej, Orzechowska & Szymańska, 2024; Zhang, Wang & Rui, 2025). The hydrodynamic size of nanoparticles refers to the effective diameter of particles as they move through a fluid medium (Casiano-Muñiz et al., 2024). When considered for germination applications, the hydrodynamic size of nanoparticles is an important feature to be determined since it provides insights into their capability to penetrate plant tissues and cell membranes, as well as nutrient delivery capacity, and possible interactions during plant growth. Here, it was noted that the hydrodynamic size of nTiO2 varied disproportionately according to the performed dilution (0–208.8 µM nTiO2; Fig. 1). The ζ-potential of nanoparticles is associated with the degree of electrostatic repulsion between adjacent, similarly-charged particles in suspension (Kamble et al., 2022). The ζ-potential of nanoparticles can be positive or negative, where higher values indicate desirable stability and lower values suggest the tendency of nanoparticles to aggregate in the medium. Similar to the hydrodynamic size, the ζ-potential of nanoparticles yields data about their capability to act as growth-promoting agents in light of their distribution and bioavailability. Considering the determined ζ-potential values of nTiO2, it is noteworthy to mention that they are prone to poor stability and low interactions with biological components.

It is well known that seed moisture content is a crucial factor in seed physiology and storage, influencing germination, longevity, and overall seed quality (Tianshun et al., 2024). In this study, it was observed that seeds imbibed with 208.8 µM nTiO2 exhibited lower weight gain compared to those treated with 104.4 µM nTiO2, with a significant difference between them (Fig. 2). In pea (Pium sativum L.), a blockage of water absorption and dehydration of seed tissues were observed after treatment with 626 µM nTiO2, with 10, 36, and 43% decreases in water content in the embryonic axes, coat, and cotyledon, respectively, after five days of treatment as compared to the control, which was attributed to a pronounced lipid peroxidation activity that disrupted the processes of transport in the membrane of water and solutes (Basahi, 2021).

As for total germination percentage (TGP; Table 1), there were no significant differences among treatments tested, though the application of 208.8 µM nTiO2 resulted in the greatest mean (96.6%). These results coincide with those observed in cucumber (Cucumis sativus L.) (Mushtaq, 2011; Andersen et al., 2016), canola (Brassica napus L.) (Mahmoodzadeh, Nabavi & Kashefi, 2013), tobacco (Nicotiana tabacum L.) (Frazier, Burklew & Zhang, 2014), and maize (Zea mays L.) (Karunakaran et al., 2016).

The germination speed coefficient (GSC) plays an important role when establishing a crop in the field, since, through this variable, a seed can be classified as fast or slow germinating (Navarro, Febles & Herrera, 2015; Talská et al., 2020). The GSC is the reciprocal of the time average of a seed to germinate and independent of the percentage of total germination, and its value ranges from 0 (no germination) to 1 (rapid germination) (Sobarzo-Bernal et al., 2021). The doses of Ti used in this study did not show significant effects on the mean values of GSC, though they tended to decrease as the Ti dose increased.

The values above the control observed in vigor index I in this study agree with what was found in faba bean (Vicia faba L.) treated with 626 µM nTiO2 (<100 nm; Sigma Aldrich, Germany) (Ruffini et al., 2016). On the contrary, higher doses (18.8 mM n TiO2; 20 nm) decrease the vigor index of canola seedlings (Mahmoodzadeh, Nabavi & Kashefi, 2013). It is likely that the optimal dosage of nTiO2 may improve germination and seed vigor because it stimulates the translocation of nutrients and activates antioxidant enzymes that reduce the accumulation of ROS causing oxidative stress (Chandoliya et al., 2024).

The percentages obtained in germination help determine the vigor of the seeds. Plant species with germination between 60 and 80% are considered to have intermediate vigor, while values greater than 80% are considered to have high vigor (Jafari, Kordrostami & Ghasemi-Soloklui, 2024). Consequently, the results shown in TGP indicate that the seeds have a high level of vigor. In addition to the high values obtained with the 156.6 and 208.8 µM nTiO2 treatments in vigor index I, it was observed that the evaluated seeds have a potential for rapid emergence, which allows obtaining healthy and vigorous seedlings under adverse environmental conditions (Navarro, Febles & Herrera, 2015), thanks to the use of Ti.

Optimal doses of nTiO2 may improve growth and differentiation processes in plants, since they induce the transcriptional activation of genes involved in photosynthesis and light capture in chloroplasts, which in turn allows better CO2 fixation, sugar biosynthesis, and biomass accumulation (Chen et al., 2024). In tomato, Pérez-Velasco et al. (2023) demonstrated that covered rutile-TiO2 nanoparticles enhance yield and growth by modulating gas exchange and nutrient status. In our study, greater root length was observed with high doses (i.e., 156.6 and 208.8 µM nTiO2) (Table 2), aligning with what was reported in cucumber and onion (Andersen et al., 2016) exposed to 12.52 µM nTiO2 (22 to 25 nm in size), and Arabidopsis thaliana seeds treated with 3.13, 6.26 and 12.52 mM nTiO2 (25 nm in size) (Szymanska et al., 2016). On the contrary, tomato seedlings treated with 0.626 to 62.6 mM Ti did not significantly increase germination percentage or root length, but an increase in SOD activity was observed when a high concentration of Ti (12.52 mM nTiO2) was applied (Song et al., 2013). Likewise, under our experimental conditions, an increase of 16.3% in stem length was observed with respect to the control when applying the 156.6 µM nTiO2 dose, as compared to the control. Under our experimental conditions, nano-TiO2 had no significant effects on the number of lateral or secondary roots; however, in the treatments with 52.2 and 104.4 µM nTiO2, a significant decrease of 32.2 and 37.1%, respectively, in the number of leaves was observed, compared to the control. The application of nTiO2 may induce autophagy in chloroplasts through oxidative stress, damaging photosynthetic activity and carbon fixation in the plant (Shull, Kurepa & Smalle, 2019), which ultimately may affect biomass production and growth.

As compared to the control, the application of 156.6 or 208.8 µM nTiO2 increased moisture content in roots (Fig. 3A), while applying 104.4 µM nTiO2 increased this variable in stems (Fig. 3B). In leaves, the highest dose (i.e., 208.8 µM nTiO2) resulted in a 2.6% increase as compared to the control, while the application of 104.4 µM nTiO2 also increased this variable as compared to the control, though at lower extent (Fig. 3C). This is a benefit for crops, because plant growth is mediated by the amount of water absorbed by the root and by physical properties such as pressure potential or turgor at the cellular level, which trigger cell elongation and expansion (Feng et al., 2016).

The increase in biomass and yield are highly dependent on the stimulation of chlorophyll biosynthesis, enzymatic activity, and the increase in photosynthetic capacity promoted by Ti (Vatankhah et al., 2023).

In our study, this effect became evident when evaluating fresh biomass (Table 3), since the application of 208.8 µM nTiO2 resulted in the greatest amount of biomass in roots and leaves (35.2 and 12.8%, respectively, as compared to the control). However, the application of low doses (52.2 and 104.4 µM nTiO2) reduced stem fresh biomass by 17.3 and 31.3%, respectively. Coincidentally, tomato seedlings treated with doses equal to or greater than 0.626 mM nTiO2 decrease the weight of fresh biomass (Song et al., 2013).

The application of 104.4 µM nTiO2 decreased the dry biomass of the stems by 43.2% (Table 4), whereas this dose reduced the dry biomass of the leaves by 27.1%, as compared to the control. A similar result was observed in total biomass of the seedling. Coincidentally, Feizi et al. (2013) reported that the application of 0.5 mM nTiO2 to fennel decreases the dry biomass weight of the roots, stems, and seedlings.

In relation to the hormetic effect (Fig. 4) in the root (RL) and stem length (SL) variables, it was observed that nTiO2 exerts different effects according to the curves obtained: inverted U-shaped for SL and J-shaped for RL. In the SL variable, the dose with 104.4 µM nTiO2 remained above the hormetic zone, while the 156.6 µM nTiO2 dose caused the highest stimulation value. Consequently, the stimulant response curve was observed at medium doses, although in other studies the response has been observed at low doses (Trela-Makowej, Orzechowska & Szymańska, 2024). When applying 208.8 µM nTiO2, toxicity was observed at the threshold. For RL, it was observed that the 104.4 µM nTiO2 dose describes a phenomenon associated with biological dysfunction or toxic damage, while higher doses (i.e., 156.6 and 208.8 µM nTiO2) show stimulation on this variable (Cox et al., 2016; Guzmán-Báez et al., 2021).

In order to perform fair comparisons among different studies, key factors have to be taken into consideration. In addition to the experimental conditions in which the studies were carried out, other factors determining the final effect of the nanoparticles include: concentration, particle size, distribution, morphological shape, chemical composition, surface characteristics, and coating, as well as the plant species (López-Herrera et al., 2024; Singh et al., 2024; Zhang, Wang & Rui, 2025). For instance, while Song et al. (2013) and Acosta-Slane et al. (2024) exposed seeds for 48 to 120 h to nTiO2 (27 nm in size; Evonik), at concentrations between 0.626 and 62.6 mM, our seeds were exposed for only 24 h to nTiO2 of <25 nm in size (Sigma-Aldrich, Germany), at concentrations between 2.5 and 10 mg L−1. When exposing tomato seeds to up to 9.393 mM nTiO2 of 25 nm in size for 1 h, Raliya et al. (2015) found no effect on germination, but aerosol-mediated applications were found to be more effective than the soil-mediated applications on the uptake of the nanoparticles and the stimulation of growth responses. Such different findings exploited different properties of nanoparticles with different doses, exposure times and concentrations (Chahardoli et al., 2022).

The effects of nTiO2 on germination, growth and development do not always relate to the exposure concentration, indicating that a mass-based concentration may not fully explain this phenomenon (Andersen et al., 2016). Experiments performed in greenhouses may have different results from those carried out in a laboratory. Nano-TiO2 displays a unique reaction to UV, such as oxidation (Lu et al., 2011), and therefore sunlight may change the toxicity of nano-TiO2. Alternatively, high temperatures in the greenhouse (45 °C) may increase the phytotoxicity of nTiO2. Overall, beneficial, neutral, or toxic effects of nTiO2 in natural conditions, subject to multiple potentially synergistic or antagonist effects, may vary in a greater manner than those demonstrated in laboratory experiments (Song et al., 2013). Hence, the fundamental mechanism behind the effects of nanoparticles on plant physiology and metabolism remains an open question. These facts may explain, at least in part, the different results observed among these approaches.

Nano-TiO2 has been claimed to trigger hormetic dose–response curves in plants and other organisms (i.e., Chandoliya et al., 2024; Trela-Makowej, Orzechowska & Szymańska, 2024). Nevertheless, no one has statistically estimated and demonstrated such hormetic effects supported with validated mathematical models. To the best of our knowledge, this study represents the first attempt to provide an in-depth statistical evaluation of hormetic dose–response curves triggered by nTiO2 in crop species of such importance as tomato.

Since Ti can promote vigor index, plant growth, and moisture content in plant tissues, as demonstrated in this study, it may provide applications for sustainable agriculture, food safety, and security.

Conclusions

We have herein demonstrated that nTiO2 can improve some parameters of seed germination and initial vegetative growth of tomato seedlings. In the range of 156.6 to 208.8 µM nTiO2, it stimulated the vigor index of seeds, root and stem length, and the moisture content in leaves, roots and the whole seedlings. Importantly, Ti increased fresh biomass weight in roots and leaves, and the effects in root and stem length display hormetic dose–response curves. Nonetheless, the highest dose (i.e., 208.8 µM nTiO2) decreased seed weight after imbibition as compared to seeds receiving 104.4 µM nTiO2, while applying 104.4 µM nTiO2 decreased dry biomass weight of stems, leaves and the whole seedling, as compares to the control. Therefore, imbibition of tomato seeds with 156.6 to 208.8 µM nTiO2 can help promote and stimulate vigor index I during germination, as well as some variables of initial growth of tomato seedlings as compared to the control. Nevertheless, contrary to our original hypothesis, for most variables measured, we found no evidence of a hormetic dose–response within the range of doses that we tested.

Supplemental Information

Supplemental Information 1 Raw data of the dependent variables measured to test the effect of titanium on germination and initial growth of tomato

Supplemental Information 2 Statistical analyses of data of the dependent variables measured to test the effect of titanium on germination of tomato seeds

Supplemental Information 3 Statistical analyses of data of the dependent variables measured to test the effect of titanium on seed weight increase of tomato

Supplemental Information 4 Statistical analyses of data of the dependent variables measured to test the effect of titanium on water content increase of tomato seeds

Supplemental Information 5 Statistical analyses of data of the dependent variables measured to test the effect of titanium on production of tomato seedlings

Supplemental Information 6 Second rebuttal letter

We thank the support staff of the Laboratory of Plant Nutrition of the College of Postgraduates in Agricultural Sciences Montecillo Campus for their help with sample collection and processing.

Additional Information and Declarations

Competing Interests

Author Contributions

Data Availability

The authors declare there are no competing interests.

Víctor Hugo Carbajal-Vázquez performed the experiments, analyzed the data, prepared figures and/or tables, authored or reviewed drafts of the article, and approved the final draft.

Libia Iris Trejo-Téllez conceived and designed the experiments, analyzed the data, prepared figures and/or tables, authored or reviewed drafts of the article, and approved the final draft.

Jorge Luis Mejía-Méndez performed the experiments, analyzed the data, prepared figures and/or tables, and approved the final draft.

Josafhat Salinas-Ruiz analyzed the data, prepared figures and/or tables, and approved the final draft.

Fernando Carlos Gómez-Merino conceived and designed the experiments, authored or reviewed drafts of the article, and approved the final draft.

The following information was supplied regarding data availability:

The raw measurements are available in the Supplementary File.

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
