# Peer review of "Biostimulant effects of titanium dioxide nanoparticles on germination and initial growth of tomato: evidence of hormesis"

_PeerJ, doi:10.7717/peerj.20516_

## Round 0.1 · original submission · Major Revisions

· Academic Editor

Major Revisions

**Language Note:** PeerJ staff have identified that the English language needs to be improved. When you prepare your next revision, please either (i) have a colleague who is proficient in English and familiar with the subject matter review your manuscript, or (ii) contact a professional editing service to review your manuscript. PeerJ can provide language editing services - you can contact us at [email protected] for pricing (be sure to provide your manuscript number and title). – PeerJ Staff

·

Basic reporting

Dear Authors,
Your manuscript addresses an important field; however, I have identified several areas that require attention. Detailed comments and suggestions have been provided in the attached PDF file. Please review them thoroughly and revise your manuscript accordingly.

Experimental design

-

Validity of the findings

-

Reviewer 2 ·

Basic reporting

The manuscript has a clear presentation and valuable results. Figures and tables are good designed.

Experimental design

This part has correctly designed and presented in detail.

Validity of the findings

Statistical analysis are enough to evaluate the data.

Reviewer 3 ·

Basic reporting

I think that in general it is good, you can clearly understand the study and the results found, as well as the conclusions. However, I suggest improving some general aspects, such as

In the title of the paper, I suggest leaving only one verb.

In paragraphs 44-58, I suggest summarising the information.

I suggest deleting the paragraphs that refer to climate change, as I believe that it is a punctual study and does not establish a direct link with the phenomenon.

I suggest standardising the units of concentration to make them comparable, especially in the discussion. For example, in lines 268 and 271.

Experimental design

I think that the most important materials and methods are available, but in NPs work there are cuestions that need to be answered, such as size, shape, Z-potential as a measure of the agglomerate state of the nanomaterial used. This information will make the study more reliable given the particular conditions of suspension and application of TiO2.

Line 125. Please provide more information about the NPs suspension and how the authors have ensured that their agglomeration is minimised.

Line 155 and 156. Please clarify whether Vigor Index I and II are given by root+stem or root+shoot, as stem does not include leaf.

Line 168. Total water content (TWC), I suggest a change to relative water content as it is a ratio and not a content.

Lines 246 to 251. I suggest deleting.

Line 305. I suggest add scientific names

Validity of the findings

In general, I think the data are valid. However, I consider that the following comments should be taken into account:

Lines 263 - 266. What is stated is an assumption, I suggest more support for what is stated. Also, what is stated contradicts what is presented in lines 268 - 269.

I think that the hormetic effect should be treated with more caution, because although this effect is observed for root and stem length parameters, it seems to be different globally, as several parameters had their best response at the highest dose.

---

## Round 0.2 · Minor Revisions

· Academic Editor

Minor Revisions

Dear Dr. Gómez-Merino, I ask you to make some minor corrections to the manuscript before the article can be approved for publication.

·

Basic reporting

All points have been addressed for the study titled “Biostimulant effects of titanium dioxide nanoparticles on germination and initial growth of tomato: Evidence of hormesis.” However, I still have two questions:

First: The author Jorge Luis Mejía-Méndez was not mentioned in the earlier version of the manuscript. Could you please clarify this change?

Experimental design

Second:
In the Methods section (lines 214–215), the study states:
“Titanium dioxide nanoparticles (nTiO₂, 99.7%, Sigma-Aldrich, <25 nm) with a concentration of 208.8 µM nTiO₂ was prepared.”
The expression seems inconsistent. Is the material described as nanoparticles with a size of <25 nm, or is it referring to a microscale concentration of 208.8 µM nTiO₂? Please verify and clarify this point.

Validity of the findings

-

---

## Round 0.3 · Minor Revisions

· Academic Editor

Minor Revisions

Dear Dr. Gómez-Merino, I kindly request that you make minor corrections to the manuscript in accordance with the reviewer's recommendations. Table 3 is a combination of two tables and cannot be published as is. Figures 2 and 3 should be presented as a box plot (median, first and third quartiles, minimum and maximum values). The number of replicates should be added in parentheses to the table and figure titles, for example, n = 10.

·

Basic reporting

Answered

Experimental design

Answered

Validity of the findings

Answered

Additional comments

Answered

Reviewer 3 ·

Basic reporting

No comments.

Experimental design

No comments.

Validity of the findings

No comments.

Additional comments

I am grateful that the authors made a great effort to address most of the comments. However, I believe two elements could be improved:
Firstly, the abstract should provide a brief overview of the conditions for plant growth and development.
Secondly, I maintain that incorporating climate change and its effects into this article would not strengthen it, given that it is a specific issue which does not directly address any of the effects of this global phenomenon. I therefore suggest deleting the paragraph between lines 406 and 409.

---

## Round 0.4 · Major Revisions

· Academic Editor

Major Revisions

I ask you to carefully review your manuscript.

According to the Conclusions (L 421) " the highest dose decreases seed weight after imbibition, as well as the number of leaves and leaf dry biomass weight."; however this conclusion seems to be false. According to table 2, the highest dose INCREASED the number of leaves (relative to 104.4) ; while for leaf dry biomass (Table 4) there is no statistical difference between the highest does and any of the other doses. Also, if the authors intend that the comparison be made only with controls then that needs to be specified in the sentence, otherwise the wording is ambiguous and may be taken to mean in comparison to all other treatments.

Additionally, the authors should clarify that contrary to their original hypothesis, for most variables measured, they found NO evidence of a hormetic dose-response within the range of doses that they tested.

Conclusions L 422 "imbibition of tomato seeds with 156.6 to 208.8 µM nTiO2 supplied as nTiO2 can help promote seed germination and stimulate initial growth of tomato seedlings" -- this statement does not seem to be supported by the Results. According to Table 1 there is no difference among any of the dosages in germination percent and speed; thus seed germination was not promoted. Vigor indices at 156.6 to 208.8 µM do not differ statistically from 52.2 µM (Table 1).

**PeerJ Staff Note**: Please ensure that all review, editorial, and staff comments are addressed in a response letter and that any edits or clarifications mentioned in the letter are also inserted into the revised manuscript where appropriate.

---

## Round 0.5 · accepted · Accept

· Academic Editor

Accept

The article can be recommended for publication.